# Temporal requirements of SKN-1/NRF as a regulator of lifespan and proteostasis in *Caenorhabditis elegans*

**Danielle Grushko**[☯], **Hana Boocholez**[☯], **Amir Levine, Ehud Cohen**[iD]*

Department of Biochemistry and Molecular Biology, The Institute for Medical Research Israel–Canada (IMRIC), The Hebrew University School of Medicine, Jerusalem, Israel

☯ These authors contributed equally to this work.
* ehudc@ekmd.huji.ac.il

## Abstract

Lowering the activity of the Insulin/IGF-1 Signaling (IIS) cascade results in elevated stress resistance, enhanced protein homeostasis (proteostasis) and extended lifespan of worms, flies and mice. In the nematode *Caenorhabditis elegans* (*C. elegans*), the longevity phenotype that stems from IIS reduction is entirely dependent upon the activities of a subset of transcription factors including the Forkhead factor DAF-16/FOXO (DAF-16), Heat Shock Factor-1 (HSF-1), SKiNhead/Nrf (SKN-1) and ParaQuat Methylviologen responsive (PQM-1). While DAF-16 determines lifespan exclusively during early adulthood and governs proteostasis in early adulthood and midlife, HSF-1 executes these functions foremost during development. Despite the central roles of SKN-1 as a regulator of lifespan and proteostasis, the temporal requirements of this transcription factor were unknown. Here we employed conditional knockdown techniques and discovered that in *C. elegans*, SKN-1 is primarily important for longevity and proteostasis during late larval development through early adulthood. Our findings indicate that events that occur during late larval developmental through early adulthood affect lifespan and proteostasis and suggest that subsequent to HSF-1, SKN-1 sets the conditions, partially overlapping temporally with DAF-16, that enable IIS reduction to promote longevity and proteostasis. Our findings raise the intriguing possibility that HSF-1, SKN-1 and DAF-16 function in a coordinated and sequential manner to promote healthy aging.

**Data Availability Statement:** All relevant data are within the manuscript and its Supporting Information files.

## Introduction

For decades, aging was thought to be an entirely stochastic, uncontrolled process driven by the accumulation of cellular damage [1,2]. This view has changed as it became evident that manipulating the activities of several genetic and metabolic pathways elevates stress resistance, enhances protein homeostasis (proteostasis) and extends lifespans of various organisms. Dietary restriction (DR) [3], reducing Insulin/IGF-1 signaling (IIS) [4], lowering the activity of the mitochondrial electron transport chain (ETC) [5] and of signaling that emanates from the

**Funding:** This study was supported by the Israel Science Foundation (ISF#981/16) awarded to EC (isf.org.il). The Israeli Ministry of Science and Technology (MOST#80884) awarded to EC (https://www.gov.il/en/Departments/ministry_of_science_and_technology) Henri J. and Erna D. Leir Chair for Research in Neurodegenerative Diseases - awarded to EC. The funders had no role in study design, data collection and analysis, decision to publish, or preparation of the manuscript.

**Competing interests:** The authors have declared that no competing interests exist.

reproductive system [6], were all found to slow the aging process. The IIS, probably the most prominent aging-regulating pathway, is a key regulator of development, stress resistance, metabolism and longevity of various organisms [4,7–9].

In the nematode *Caenorhabditis elegans* (*C. elegans*), upon binding of one of its ligands, the lone insulin/IGF-1 receptor DAF-2 activates a signaling cascade which regulates the activity of a nexus of transcription factors through a highly conserved set of molecular components. DAF-2's downstream kinases mediate the phosphorylation of the transcription factors DAF-16 [10,11] and SKN-1 [12]. These phosphorylation events retain DAF-16 and SKN-1 in the cytosol, preventing them from regulating their target gene networks. Analogously, the IIS negatively regulates the activity of HSF-1 by preventing the phosphorylation of DDL-1, a protein that interacts with this transcription factor. Non-phosphorylated DDL-1 along with DDL-2 and HSB-1 form a complex of proteins that binds HSF-1 and retains it in the cytosol [13]. The IIS also governs the cellular localization of PQM-1, a transcription factor which responds to IIS reduction in opposition to DAF-16, and plays key roles in the IIS-controlled lifespan determining mechanism [14]. Thus, knocking down the activity of DAF-2 by either mutation or RNA interference (RNAi) hyper-activates HSF-1, DAF-16 and SKN-1, creating long-lived, youthful and stress-resistant worms. These longevity and stress resistance effects of *daf-2* knockdown are dependent upon each of the aforementioned transcription factors [12,15,16]. Similarly to worms, reduced IGF-1 signaling was shown to extend the lifespan of mice [9], and mutations in components of the same pathway correlate with extreme longevity of humans [7,17], indicating that the aging-regulating roles of the IIS are conserved from worms to mammals.

## The alteration of aging protects worms and mammals from toxic protein aggregation

Maintaining the integrity of the proteome is vital for organismal functionality and viability. However, as an organism ages, its ability to maintain proteostasis declines [18,19], enabling subsets of proteins to form potentially toxic aggregates that accrue within the cell [20]. In some cases, the accumulation of aggregated proteins underlies the development of a myriad of late-onset maladies including neurodegenerative disorders such as Alzheimer's disease (AD) [21] and Huntington's disease (HD) [22]. Aging is the major risk factor for the manifestation of neurodegeneration, a common feature in these late-onset diseases [23]. This raises the prospect that the alteration of aging could maintain proteostasis in the late stages of life thereby preventing, or at least delaying, the emergence of neurodegeneration. Indeed, IIS reduction [24–26], DR [27], ETC impairments [28] and germ cell ablation [29], were all found to promote proteostasis and protect model nematodes from toxic protein aggregation (proteotoxicity). These mechanistic links define proteostasis collapse as an inherent aspect of aging [30]. Importantly, all the aforementioned IIS regulated transcription factors; DAF-16, HSF-1 [24,25], SKN-1 [31] and PQM-1 [32] are involved in the regulation of proteostasis, raising the prospect that modulating the activities of these factors could extend healthspan through late stages of life. However, to maintain proteostasis and extend healthspan without affecting lifespan, it is critical to ascertain the temporal requirements of these factors as lifespan and proteostasis regulators.

## The temporal requirements of DAF-16 and HSF-1

Reducing the IIS at different stages of life via *daf-2* RNAi, identified that IIS reduction during reproductive adulthood (days 1–6) and no other stage of life, extends the lifespan of *C. elegans* [33]. Consistently, an increase in lifespan was observed in the fruit fly *Drosophila melanogaster* when dFOXO (the ortholog of DAF-16) was over-expressed during reproductive adulthood

but not during any other stages of life [8]. Surprisingly, we discovered that HSF-1 is of foremost importance for the determination of lifespan during the L2 larval stage, but also has a marginal effect on lifespan during reproductive adulthood [34]. DAF-16 and HSF-1 also exhibit distinct temporal requirements for proteostasis maintenance. While DAF-16 is dispensable for proteotoxicity protection during development and plays its counter-proteotoxic protective roles exclusively during adulthood, HSF-1 executes these functions mainly during development [35]. These distinct temporal patterns raise questions about the functional relationship between these two transcription factors, SKN-1 and the IIS.

Despite the central roles of SKN-1 as a regulator of lifespan downstream of the IIS [12] and via the DR pathway [36], as well as its influence on proteostasis [31], the temporal requirements of SKN-1 for these functions were unknown. To address this, we used the nematode *C. elegans* and a conditional RNAi knockdown technique and found that SKN-1 governs lifespan and proteostasis primarily during late development and early adulthood.

## Materials and methods

### Worm and RNAi strains

N2 (wild type, Bristol), CB1370 (*daf-2* (e1370)), CL2006 (*unc-54*p::human Aβ$_{3-42}$), CF512 (fer-15(b26)II; fem-1(hc17)IV) and DA1116 (*eat-2* mutant) worms were obtained from the Caenorhabditis Genetics Center (CGC, Minneapolis, MN), which is funded by the National Institutes of Health Office of Research Infrastructure Programs (P40 OD010440). AGD1246 (*rgef-1p*::Aβ$_{1-42}$; RF4 rol-6) worms were a generous gift of Dr. Andrew Dillin (University of California at Berkeley). AM140 (rmIs141[unc-54p::Q35::YFP]) and AM1126 (rmIs383[Pf25b3.3::Q(35)::YFP]) worms were a generous gift of Dr. Richard Morimoto (Northwestern University). All worm strains were routinely grown at 15˚C for maintenance. For experimentation, all worms were kept at 20˚C throughout life except for CF512 animals which are heat-sensitive feminized and were therefore grown at 25˚C throughout development to prevent progeny, and then maintained at 20˚C throughout adulthood. To reduce gene expression, we used bacterial strains expressing dsRNA: empty vector (pAD12), *skn-1* and *dcr-1* dsRNA expressing bacteria from the M. Vidal RNAi library. Each RNAi bacteria colony was grown at 37˚C in LB with 100μg/ml ampicillin and then seeded on NG-ampicillin plates and supplemented with 100mM Isopropyl β-D-1-thiogalactopyranoside (IPTG 1mM final concentration).

### Expression analysis by quantitative real-time PCR (qPCR)

Synchronized eggs were placed on NGM plates seeded with the indicated bacteria. The worms were grown from hatching until day one of adulthood unless otherwise indicated. The worm samples were then harvested and washed with M9 buffer to remove bacteria from the samples. Each worm pellet was re-suspended in 1M DTT and RA1 (solution from the NucleoSpin® RNA kit (Macherey-Nagel, Duren Germany #740955.50)) and frozen at -80˚C overnight. After thawing the samples on ice, zirconium oxide beads (Next Advance, ZrB05) were added to the samples and the samples were homogenized at 4˚C using a Bullet Blender® (Next Advance). To separate RNA from protein and other materials, samples underwent centrifugation at room temperature in a tabletop centrifuge. The NucleoSpin RNA isolation Kit (Macherey Nagel, Duren Germany #740955.50) was used according to the manufacturer instructions to extract RNA. cDNA was generated by reverse transcription of the total RNA samples with iScriptRT Advanced cDNA Synthesis Kit for RT-PCR (Bio-Rad, Hercules, CA; #170–8891;). qPCR was performed in triplicates using the iTaq™ Universal SYBR® Supermix (Bio-Rad; #172–5124) and quantified in a CFX96™ Real-Time PCR Detection System (Bio-Rad). The levels were normalized to the levels of *cdc-42*, *act-1* and/or *pmp-3* cDNA.

| Primer name | Forward sequence | Reverse sequence |
|---|---|---|
| *act-1* | GAG CAC GGT ATC GTC ACC AA | TGT GAT GCC AGA TCT TCT CCA T |
| *pmp-3* | GTT CCC GTG TTC ATC ACT CAT | ACA CCG TCG AGA AGC TGT AGA |
| *cdc-42* | CTG CTG GAC AGG AAG ATT ACG | CTC GGA CAT TCT CGA ATG AAG |
| *skn-1* | CGA GAT CGT TCA TAT TCA AGC | CAC ATA CTG GCC AGA TGG |

## Lifespan assays

Lifespan assays were conducted as previously described [34]. Briefly, synchronized eggs were placed on master NG-ampicillin 9cm plates seeded with the indicated RNAi bacterial strain and supplemented with 100mM IPTG (~1mM final concentration). After synchronization by bleach solution (0.75N KOH, 1.8% hypochlorite), the worms were places on RNAi plates as indicated. L2 larvae were collected 28 hours after bleach and L4 larvae 48 hours. The developmental stages of larvae were validated by visualization by light microscopy. CF512 worms were grown at 25˚C throughout development to avoid progeny, then transferred to 20˚C for the duration of their life. *daf-2* (e1370) mutant animals (strain CB1370) as well as N2 and DA1116 worms were developed and maintained at 20˚C. At day 1 of adulthood, 120 animals per treatment were transferred onto 5cm NG-ampicillin plates (12 animals per plate). Worms that failed to move their heads when tapped twice with a platinum wire or when a hot pick was placed proximally to their body were scored as dead. Survival rates were recorded daily.

## Proteotoxicity assays

To follow Aβ-mediated toxicity by the "paralysis assay" [25], synchronized CL2006 or AGD1246 worms were grown on NG plates containing 100μg/ml ampicillin, spotted with E. coli cultures that express dsRNA as indicated. On day one of adulthood, 120 worms were transferred onto 10 5mm NG-ampicillin plates (12 animals per plate). These 10 plates were randomly divided into 5 sets (2 plates, 24 worms per set) to prevent potential bias. Paralysis of these worms was scored daily by gently tapping their noses with a platinum wire or placing a hot pick proximally to their bodies. Worms that were capable of moving their noses but unable to move the trunk of their bodies were scored as "paralyzed" and removed from the plates. The assay was terminated at day 12 or 13 of adulthood in order to avoid scoring old animals as paralyzed. As a control, this assay was also performed using wild type N2 worms.

To follow the toxicity of polyQ35-YFP stretches by the "thrashing assay" [37], synchronized eggs of AM140 or AM1126 worms were placed on plates seeded with control bacteria (EV) or bacteria that express RNAi towards *skn-1*. At the indicated ages, one worm was placed in a 10μL drop of M9 buffer and the number of body bends per 30 seconds was scored. At each time point at least 20 animals were used. As a control, this assay was also performed using wild type N2 worms.

## Statistical analyses

To quantitatively measure statistical significance for the paralysis assay and thrashing assay, two-way ANOVA followed by post hoc Holm-Šídák's corrections for multiple comparisons were used since in this statistical method the assumption is that each comparison is independent of the others [31]. For qPCR experiments the statistical significance of differences was assessed using Student T-test using two-tailed distribution and two-sample equal variance. The analyses of the experiments were conducted using a minimum of three independent biological repeats of each experiment as indicated. Statistical information of lifespan experiments

is presented in the supplemental tables. All the statistical analyses and plotting of the data were performed using GraphPad Prism 9 (GraphPad Software, Inc., La Jolla, USA).

# Results

## skn-1 is involved in lifespan determination in late development and early adulthood

We sought to determine when the knockdown of *skn-1* regulates lifespan and proteostasis. To properly characterize the efficiency of conditional RNAi-mediated knockdown we took two measures. First, we tested the efficiency of *skn-1* RNAi using quantitative real-time PCR (qPCR). To avoid possible effects of developing embryos on gene expression, we used feminized CF512 worms, a strain whose lifespan is similar to that of wild type animals [38]. The worms were grown from hatching on bacteria that harbor the empty RNAi vector (EV) or cultured on *skn-1* RNAi bacteria, harvested at day 1 of adulthood and *skn-1* expression levels were measured via qPCR analysis. S1A and S1B Fig shows that *skn-1* RNAi treatment is highly efficient when applied from hatching, as it reduced the expression of the gene to less than 10% compared to untreated animals (grown on EV bacteria). This reduction was observed when three different normalizing genes, *act-1*, *pmp-3* (A) and *cdc-42* (B) were used.

Secondly, to estimate the lag time between transferring worms to *skn-1* RNAi bacteria and a notable reduction in *skn-1* expression, we examined the kinetics of *skn-1* RNAi-mediated knockdown. CF512 worms were cultured on EV bacteria and on day 1 of adulthood were transferred onto plates seeded with *skn-1* RNAi 3, 6, 9, or 12 hours prior to harvest. qPCR was used to compare the levels of *skn-1* expression to those of untreated animals. As expected, treating worms with *skn-1* RNAi from hatching resulted in an approximate 95% reduced expression. Exposure to RNAi for longer periods led to a gradual reduction that reached ~50% reduction after 12 hours (S1C Fig). An exposure of 24 hours resulted in a similar rate of ~55% reduction (S1D Fig). Therefore, exposing worms to *skn-1* RNAi requires approximately 12 hours to reach an efficient reduction in *skn-1* expression. Although the depletion of SKN-1 protein is not known, it is likely that 12 hours is sufficient to reduce the activity of this transcription factor. Thus. a lag of 12 hours in the knockdown of *skn-1* expression was considered throughout the study.

Next, we asked when SKN-1 regulates the lifespan of wild type worms (strain N2). To address this, we knocked down the expression of *skn-1* by RNAi from different stages during the worm's lifecycle. Synchronized eggs of N2 animals were placed on plates seeded with EV bacteria, or with *skn-1* RNAi expressing bacteria. At larval stages L2, L4 or at day 1, 5 or 9 of adulthood, groups of 120 worms were picked from EV plates and transferred onto *skn-1* RNAi bacteria. Lifespans were followed by daily scoring of dead animals. While worms that were grown throughout life on EV bacteria had a mean lifespan of 18.12±0.51 days (±SEM), animals that were treated from hatching with *skn-1* RNAi exhibited a significantly (p<0.001) shorter mean lifespan of 16.17±0.32 days (Fig 1A and 1B and S1 Table). This result is consistent with the previously reported shortening effect of *skn-1* mutation on the lifespan of these worms [12]. Interestingly, the knockdown of *skn-1* from the L2 or L4 larval stages resulted in similar lifespan shortening effects of 11.8% (mean lifespan 15.98±0.24 days, p<0.001) and 14.23% (mean lifespan 15.54±0.35 days, p<0.001), respectively (Fig 1A and S1 Table). These lifespans were very similar to the lifespan of animals grown on *skn-1* RNAi throughout life (mean lifespan of 16.17±0.32 days, p<0.001). Considering the 12 hours delay from the application of *skn-1* RNAi until an efficient reduction in the expression of *skn-1* is achieved and the similar lifespans of worms that were grown from hatching on *skn-1* RNAi and their counterparts that were treated from the L4 larval stage, we conclude that SKN-1 is dispensable as a regulator of

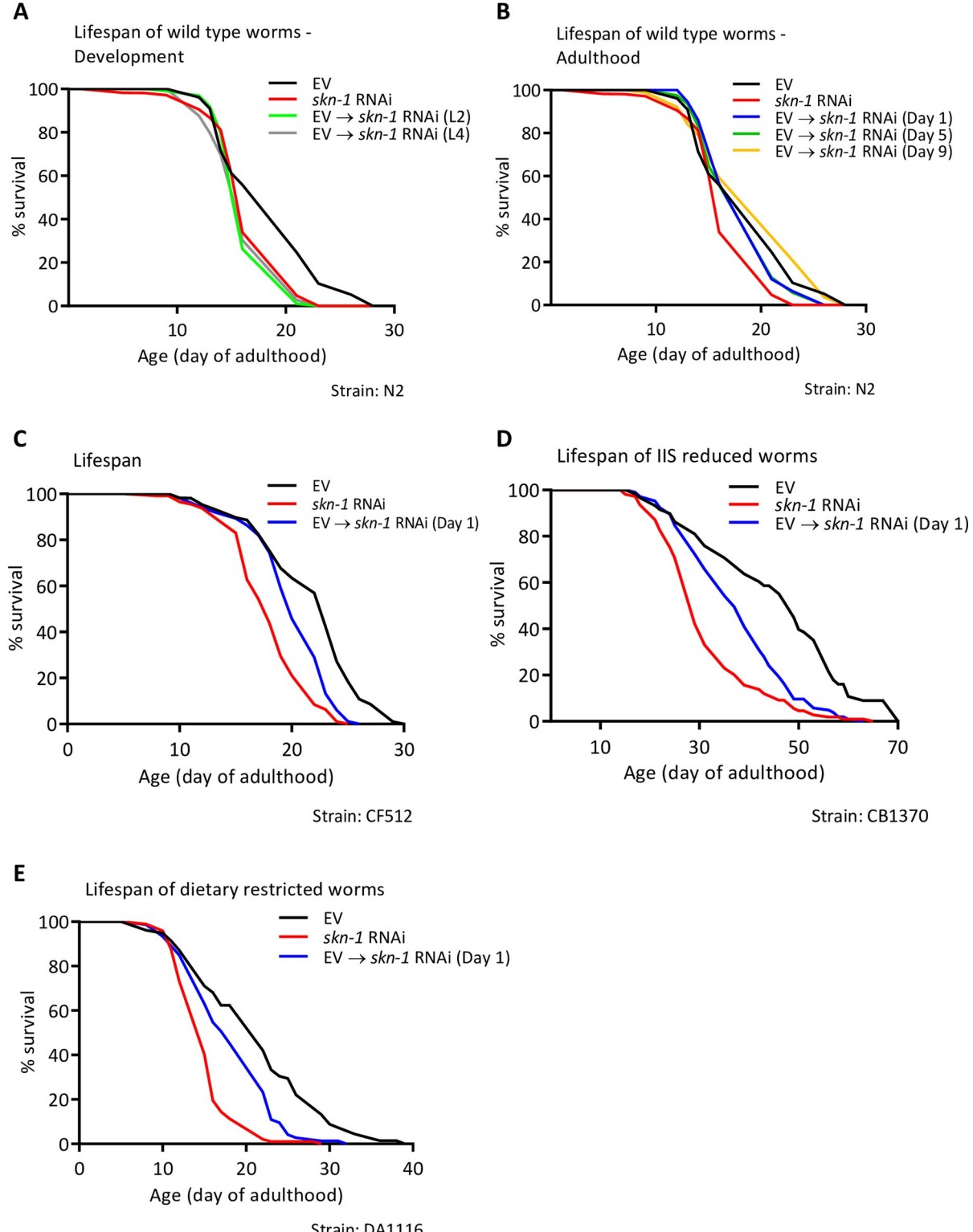

**Fig 1. *skn-1* regulates lifespan primarily from late larval development through early adulthood.** (**A-B**) The lifespan of wild type animals (WT, strain N2) treated from hatching with empty vector bacteria (EV, control), *skn-1* RNAi, or transferred from EV bacteria onto *skn-1* RNAi either during developmental stages L2 or L4 (A), or at day 1, 5, or 9 of adulthood (B) was measured. Worms treated with *skn-1* RNAi throughout life, or during developmental stages L2 or L4 showed significant reductions in lifespan (10.75%, 11.80%, and 14.23%, respectively, S1 Table). Worms treated with *skn-1* RNAi from day 1 or 5 of adulthood showed a trend of reduction in lifespan, though the observed lifespan

shortening was not significant (S1 Table). Treating worms with *skn-1* RNAi from day 9 of adulthood did not affect lifespan (S1 Table). (**C**) CF512 animals were grown throughout life on EV or *skn-1* RNAi bacteria or let hatch on EV bacteria and transferred onto *skn-1* RNAi bacteria at day 1 of adulthood. Lifespans of all groups were followed daily. The knockdown of *skn-1* from day 1 of adulthood resulted in a significant shortening of lifespan compared to untreated animals (S2A and S2B Table). (**D-E**) The knockdown of *skn-1* throughout life or from day 1 of adulthood, resulted in lifespan shortening of *daf-2* (*e1370*) mutant worms (strain CB1370, D and S3A and S3B Table) and of *eat-2* (strain DA1116) mutant animals (D and S4A and S4B Table).

lifespan during almost the entirety of larval development. In contrast, the knockdown of *skn-1* from day 1 and 5 of adulthood resulted in a trend of lifespan shortening, however, this effect was not significant (Fig 1B and S1 Table, mean lifespan 17.83±0.31 days, p = 0.31 and mean lifespan 17.51±0.42 days, p = 0.18, respectively). To further test whether SKN-1 is needed for longevity assurance during adulthood we conducted an additional set of lifespan experiments using CF512 worms. Two groups of animals were grown on EV or *skn-1* RNAi throughout life, or cultured on EV bacteria and transferred at day 1 of adulthood onto plates seeded with *skn-1* RNAi. Our results show that the lifespans of these animals are somewhat longer than these of N2 worms, however, they also indicate that the knockdown of *skn*-1 from day 1 of adulthood significantly shortened the lifespan of CF512 worms (Fig 1C and S2A and S2B Table). These results show that *skn*-1 is primarily needed later than the L4 larval developmental stage through early adulthood to regulate lifespan.

To further examine the timing requirements of SKN-1 for longevity assurance, we employed long-lived mutant worm strains. *daf-2* (*e1370*) mutant worms (strain CB1370) carry a weak *daf-2* allele and thus, exhibit exceptional longevity [4]. The animals were grown throughout life either on EV or on *skn-1* RNAi bacteria. An identical group of worms was hatched on EV bacteria and transferred onto *skn-1* RNAi bacteria at day 1 of adulthood. While the knockdown of *skn-1* from hatching resulted in lifespan shortening of 31.6% compared to the lifespan of untreated worms (mean lifespan of 30.63±0.94 and 44.79±1.97 days, respectively), knocking down the expression of this gene exclusively during adulthood shortened lifespan by only 17.34% (Fig 1D and S3A Table, mean lifespan of 37.02±1.02). These results, which were verified with an additional biological experimental repeat (S3B Table), indicate that *skn-1* is needed from day 1 of adulthood as a modulator of lifespan. Nevertheless, the observation that knocking down *skn-1* from day 1 of adulthood did not shorten lifespan as efficiently as *skn-1* RNAi treatment throughout life, suggests that this transcription factor is also needed during larval development to allow *daf-2* mutant worms to exhibit their full longevity potential. We repeated this experiment using DA1116 worms which carry a mutation in the *eat-2* gene, resulting in a pharyngeal defect that leads to constitutive dietary restriction, and thus, are long-lived [39]. We found that similarly to *skn-1* RNAi-treated *daf-2* (*e1370*) mutant worms (CB1370), the knockdown of *skn-1* throughout life shortens the lifespan of DA1116 animals by 27.89% (mean lifespan of 14.93±0.35 days). The lifespan of their counterparts who were treated with *skn-1* RNAi from day 1 of adulthood was shortened by 13.58% (mean lifespan of 17.89±0.60 days), relative to the control worms (Fig 1E and S4A Table, mean lifespan 20.70±0.90 days). These results were confirmed with an additional biological experimental repeat (S4B Table). Together, these results indicate that SKN-1 is at least partially required during developmental stages as a regulator of lifespan. However, SKN-1 is also needed during adulthood to promote the natural lifespan of wild type animals and confer the full longevity of long-lived mutant worms.

## The temporal roles of skn-1 in proteostasis maintenance

We next investigated when during the worm's lifecycle SKN-1 regulates proteostasis. To determine this, we utilized CL2006 worms which express the AD-causing amyloid beta (Aβ) peptide

in their body wall muscles [40]. This expression results in a progressive paralysis within the worm population; a phenotype that can be tracked by the "paralysis assay" via a daily scoring of paralyzed animals [25]. First, we tested whether the knockdown of *skn-1* throughout life enhances the paralysis phenotype of these animals and found that it does (Fig 2A). To confirm that the observed results stem from the counter-proteotoxic roles of *skn-1* and that its knockdown does not induce paralysis independently, we tested whether the knockdown of *skn-1* confers paralysis in wild type worms. We found that the knockdown of *skn-1* in wild type worms did not enhance the rate of aging-associated paralysis up until day 12 of adulthood (S2A Fig). To test whether the paralysis phenotype is tissue specific, we performed an identical experiment using AGD1246 worms which express the Aβ peptide under the regulation of the *rgef-1* pan-neuronal promoter [28] and found that, similarly to the observed phenotype in muscles, that the knockdown of *skn-1* by RNAi results in an increased rate of paralysis of this worm population (S2B and S2C Fig).

To establish the temporal requirement of *skn-1* as a modulator of proteostasis we treated CL2006 worms with *skn-1* RNAi throughout the experiment or from the L2 or L4 larval stages. An identical group of CL2006 worms was grown throughout the experiment (until day 12 of adulthood) on EV bacteria (see illustration in S3 Fig). Three independent experiments indicated that worms which were treated with *skn-1* RNAi throughout the experiment and their counterparts that were transferred onto *skn-1* RNAi bacteria at the L2 developmental stage, were paralyzed at similar rates, significantly higher than that of untreated animals (EV). These results indicate that *skn-1* has no role during early development (L2 stage and earlier) as a modulator of Aβ-mediated proteotoxicity. Worms that were transferred onto *skn-1* RNAi from the L4 larval stage exhibited a higher rate of paralysis than the control group (EV). However, the rate of paralysis within this worm population was lower than that of nematodes that were treated from the L2 stage (Fig 2B). This shows that even when considering the 12 hours delay in the knockdown of *skn-1* by RNAi (S1C Fig), SKN-1 activity in late developmental stages is needed for partial protection from Aβ.

To test whether *skn-1* is required during adulthood as a modulator of proteostasis, we conducted a similar experiment in which CL2006 worms were grown on EV bacteria and then transferred onto *skn-1* RNAi at either day 1, 5 or 9 of adulthood (S3 Fig). Rates of paralysis were scored daily. Three independent experiments showed that the knockdown of *skn-1* at day 1 of adulthood enhances the rate of paralysis within the population (Fig 2C). This effect, however, was less prominent than that of knocking down *skn-1* during development (Fig 2B). No significant enhancement in the paralysis phenotype, compared to untreated worms, was observed when worms were treated with *skn-1* RNAi from day 5 or 9 of adulthood (Fig 2C).

These results propose that SKN-1 is foremost required as a proteostasis regulator during late larval development through early reproductive adulthood. To further test this conclusion, we conducted a reciprocal set of experiments using *dcr-1* RNAi. DICER, encoded by *dcr-1*, is a nuclease that cleaves double stranded RNA to create small interfering RNA (siRNA) and thus, is crucial for the functionality of the RNAi machinery [41]. Accordingly, the knockdown of *dcr-1* by RNAi inactivates the RNAi machinery and partially restores the expression of the knocked down gene [33]. We utilized this technique to conditionally knockdown *skn-1* and followed the rates of paralysis of CL2006 worm populations that hatched on *skn-1* RNAi bacteria and were then transferred onto plates seeded with *dcr-1* RNAi at the L2 or L4 larval stages. Three experimental repeats indicated that the knockdown of *dcr-1* had no effect on the rate of paralysis, as animals grown on control bacteria (EV) and their counterparts that were treated with *dcr-1* RNAi throughout the experiment, had similar rates of paralysis (Fig 2D). As expected, the knockdown of *skn-1* throughout the assay increased paralysis. However, knocking down *skn-1* solely during early development, from hatching up until the L2 larval stage,

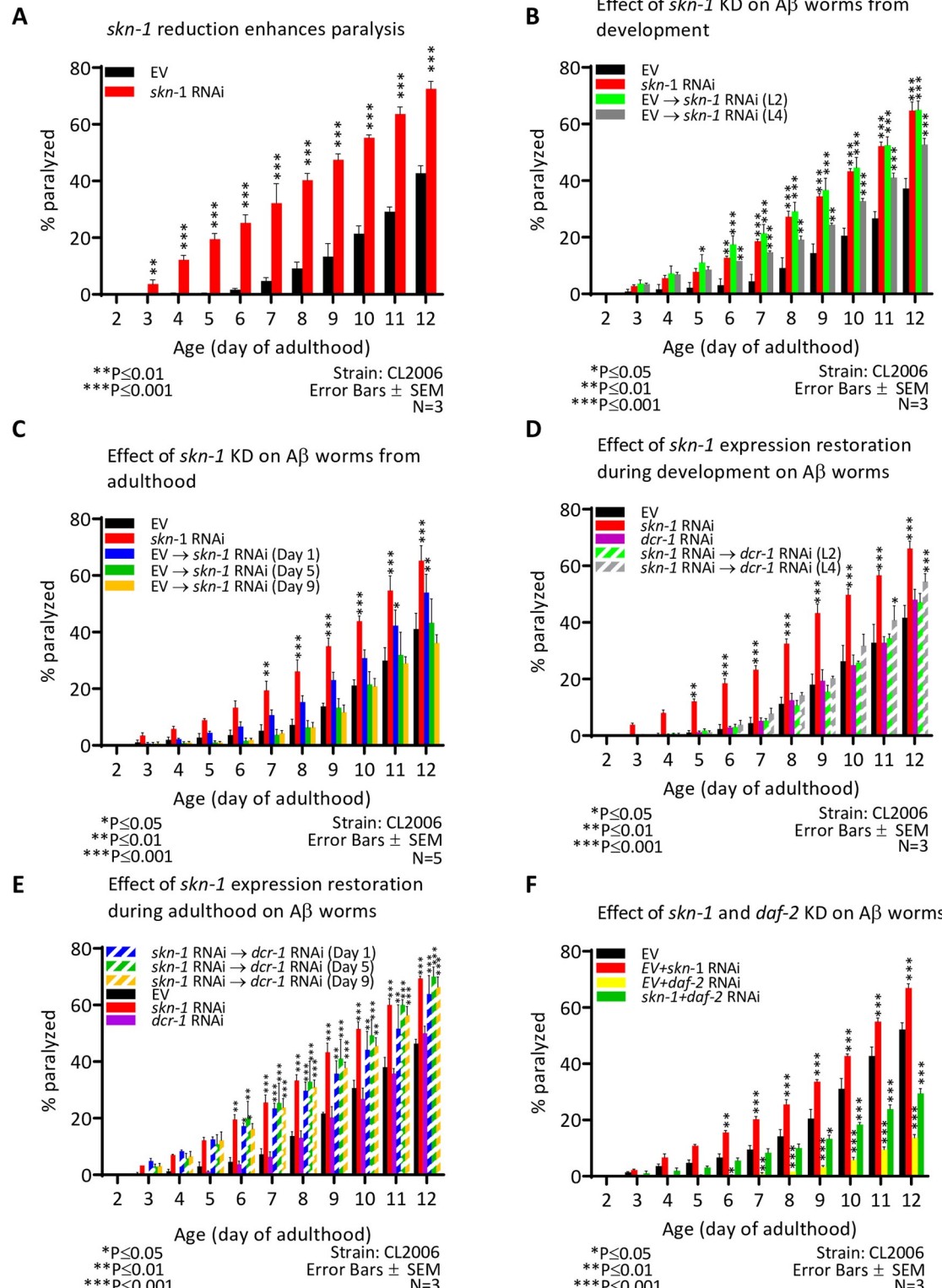

**Fig 2. *skn-1* is foremost important from late stages of larval development through day 1 of adulthood to protect against Aβ-induced proteotoxicity.** (**A**) The knockdown of *skn-1* significantly enhances the rate of paralysis of Aβ worms as observed in three independent repeats (p<0.001 see also S2C Fig). EV: *n* = 282, 78 censored; *skn-1* RNAi: *n* = 352, 8 censored. (**B**) Treating Aβ worms with *skn-1* RNAi from the L2 larval stage enhances the rate of paralysis as efficiently as lifelong treatment with *skn-1* RNAi. In contrast, knocking down *skn-1* RNAi from the L4 larval stage less pronouncedly, but significantly enhances the rate of paralysis.

EV: *n* = 269, 91 censored; *skn-1* RNAi: *n* = 338, 22 censored; EV → *skn-1* RNAi (L2): *n* = 332, 28 censored; EV → *skn-1* RNAi (L4): *n* = 342, 18 censored. (**C**) Worms grown from hatching on EV and transferred at day 1 of adulthood onto *skn-1* RNAi show a marginal increase in paralysis compared to worms grown for life on EV. In contrast, worms transferred onto *skn-1* RNAi at day 5 or 9 of adulthood show no increased rate of paralysis compared to their counterparts grown on EV throughout life. EV: *n* = 282, 78 censored; *skn-1* RNAi: *n* = 342, 18 censored; EV → *skn-1* RNAi (Day 1): *n* = 326, 34 censored; EV → *skn-1* RNAi (Day 5): *n* = 296, 65 censored; EV → *skn-1* RNAi (Day 9): *n* = 287, 73 censored. (**D**) Treating worms with *dcr-1* RNAi throughout life, or growing worms on *skn-1* RNAi from hatching and transferring them to *dcr-1* RNAi during the L2 or L4 stages of development did not enhance the rate of paralysis within the population, apart from a very slight increase at day 12 of adulthood in the population transferred at the L4 stage. EV: *n* = 279, 81 censored; *skn-1* RNAi: *n* = 320, 40 censored; *dcr-1* RNAi: *n* = 300, 60 censored; *skn-1* RNAi → *dcr-1* RNAi (L2): *n* = 313, 47 censored; *skn-1* RNAi → *dcr-1* RNAi (L4): *n* = 292, 68 censored. (**E**) Aβ worms that were grown on *skn-1* RNAi from hatching then transferred onto *dcr-1* RNAi at either day 1, 5 or 9 of adulthood and animals that were treated with *skn-1* RNAi throughout life, exhibited similarly enhanced rates of paralysis compared to the EV treated population. EV: *n* = 292, 68 censored; *skn-1* RNAi: *n* = 333, 27 censored; *dcr-1* RNAi: *n* = 300, 60 censored; *skn-1* RNAi → *dcr-1* RNAi (Day 1): *n* = 308, 52 censored; *skn-1* RNAi → *dcr-1* RNAi (Day 5): *n* = 335, 25 censored; *skn-1* RNAi → *dcr-1* RNAi (Day 9): *n* = 332, 28 censored. (**F**) CL2006 worms were grown from hatching on untreated (EV) bacteria or grown on one of the following mixtures of RNAi bacteria: EV+*skn-1*, EV+*daf-2* or *skn-1*+*daf-2*. Rates of paralysis were recorded daily. Worms that were treated with a mixture of *skn-1* RNAi+*daf-2* RNAi bacteria exhibited higher rates paralysis than their counterparts that were grown on a mixture of EV+*daf-2* RNAi. EV: *n* = 340, 58 censored; EV+*skn-1* RNAi: *n* = 350, 10 censored; EV+*daf-2* RNAi: *n* = 325, 35 censored; *daf-2* +*skn-1* RNAi: *n* = 325, 15 censored.

did not increase the rate of paralysis. The knockdown of *skn-1* from hatching up until the L4 larval stage had a small deleterious effect, as the rate of paralysis was significantly higher than that of untreated animals at days 11 and 12 of adulthood. These results suggest that SKN-1 is needed as a regulator of proteostasis from a late stage of larval development.

To further scrutinize the temporal requirements of *skn-1* as a regulator of proteostasis, we tested how the knockdown of *skn-1* affects the paralysis of Aβ worms during adulthood. Synchronized eggs were placed on plates seeded with *skn-1* RNAi bacteria and transferred onto *dcr-1* RNAi plates on either day 1, 5 or 9 of adulthood. Our results (Fig 2E) indicate that worms treated with *skn-1* RNAi throughout development and transferred onto *dcr-*1 RNAi at either day 5 or 9 of adulthood, exhibited similar rates of paralysis to animals fed with *skn-1* RNAi bacteria throughout life. Animals that were transferred onto *dicer* RNAi plates at day 1 of adulthood exhibited significantly elevated rates of paralysis within the population.

Together our results demonstrate that SKN-1 is required for protection against Aβ induced proteotoxicity from the L4 stage of larval development through the first day of adulthood. However, the restoration of *skn-1* expression at day 5 or 9 of adulthood did not rescue the enhanced paralysis phenotype, indicating that this transcription factor is dispensable as a proteostasis regulator in late stages of adulthood. These temporal requirements support the notion that SKN-1 is required for proteostasis maintenance during late developmental stages and early adulthood.

To test whether the effects of *skn-1* RNAi on the rates of paralysis stem from its roles as an IIS component, we asked whether a concomitant knockdown of *daf-2* and of *skn-1* prevent IIS reduction from providing its full protection from Aβ-mediated proteotoxicity. First, we utilized qPCR to test whether the dilution of *skn-1* RNAi with EV bacteria (50:50) affects its effectiveness. We found that the level of *skn-1* expression in worms grown on this mixture was approximately 20% compared to the levels seen in untreated worms and very similar to the level that we observed in worms that were fed solely with *skn-1* RNAi bacteria (S4 Fig). Next, we cultured Aβ worms on a mixture of *daf-2* RNAi and *skn-1* RNAi bacteria, subjected them to the paralysis assay and found that the knockdown of *skn-1* reduces the rate of the counter-proteotoxic protective effect that is conferred by *daf-2* RNAi (Fig 2F). It is important to note that since SKN-1 is activated by additional aging-regulating pathways [36], additional experiments are needed to test whether the effect of *skn-1* RNAi emanates from its role as a component of the IIS.

## The knockdown of skn-1 exacerbates proteotoxicity of polyQ35-YFP

We next sought to test whether this temporal pattern of SKN-1 as a proteostasis modulator is also true for worms that are challenged by the aggregation of a proteotoxic protein other than Aβ. To address this, we utilized worms that express a chimeric, fluorescently-tagged polygluta-mine protein of 35 repeats (polyQ35-YFP) in their body wall muscles (strain AM140). Abnor-mally long polyglutamine stretches in different proteins underlie the development of several human neurodegenerative maladies, including HD [22] and Machado Joseph Disease (MJD) [42]. AM140 animals accumulate aggregates and exhibit progressive motility impairment, [26], a phenotype that can be followed by the "thrashing assay" [37]. First, we tested whether the knockdown of *skn-1* affects polyQ35-YFP toxicity by comparing the thrashing rates of AM140 worms that were treated from hatching with *skn*-1 RNAi to those of untreated animals (EV). We found that the knockdown of *skn-1* results in a significantly reduced rate of motility on days 2, 4 and 6 of adulthood (Fig 3A), while knocking down *skn-1* in wild type worms results in only a slight reduction in motility at day 6 (S5A Fig). Similar results to those observed in the AM140 worms were obtained when thrashing experiments were conducted using worms that express polyQ35-YFP under the *rgef-1* pan-neuronal promoter (strain AM1126, S5B Fig), indicating that this phenotype is not tissue specific (In these worms, thrashing was measured at days 4 and 8 as they exhibit the proteotoxic phenotype later in life than AM140 animals). We next tested when SKN-1 protects worms from polyQ35-YFP by growing AM140 animals on EV bacteria and transferring them onto plates seeded with *skn-1* RNAi bacteria at the L2 or L4 larval stages or at day 1 or 3 of adulthood. Thrashing rates were scored at days 3 and 6 of adulthood. Our results indicate that analogously to its roles in the mitigation of Aβ proteotoxicity, SKN-1 is foremost important as a regulator of proteostasis during late larval development through early stages of adulthood (Fig 3B and 3C).

## Discussion

Our temporal analysis indicates that *skn-1* is predominantly required for lifespan determina-tion and for protection from proteotoxicity, from late stages of larval development through early adulthood (Fig 3D). Because the IIS regulates lifespan [33] and proteostasis [35] during adulthood, it is likely that during the late stages of larval development through early adulthood, SKN-1 regulates the expression of gene networks that enable IIS reduction to promote these functions later in life. This conjecture is supported by the observation that a concurrent knock-down of *daf*-2 and of *skn-1* reduces the rate of protection that emanates from IIS reduction (Fig 2F). Yet, it is also possible that SKN-1 provides protection from proteotoxicity by a differ-ent mechanism and thus, the paralysis rates that were observed in animals that were concur-rently fed with *skn-1* and *daf-2* RNAi bacteria, is the sum of the protective effect of *daf-2* RNAi and the deleterious outcome of knocking down *skn-1*. More experimental work is needed to better clarify this issue.

SKN-1 is required as a lifespan and proteostasis regulator in a time window which is subse-quent to that of HSF-1, and partially overlapping and preceding the time window in which DAF-16 executes these functions [33–35]. Interestingly, SKN-1 is also, at least partially, needed during development for DR-promoted longevity (Fig 1D). In contrast, the transcription factor PHA-4, which is also crucial for DR-mediated longevity, is solely needed during adulthood to enable this phenotype [43].

These observations substantiate that different transcription factors are needed in a sequen-tial manner during the nematode's lifecycle and raise the question of how SKN-1 acts, and what genes it regulates during late stages of development through early adulthood to enable the promotion of longevity and proteostasis in later stages of life. One possibility suggests that

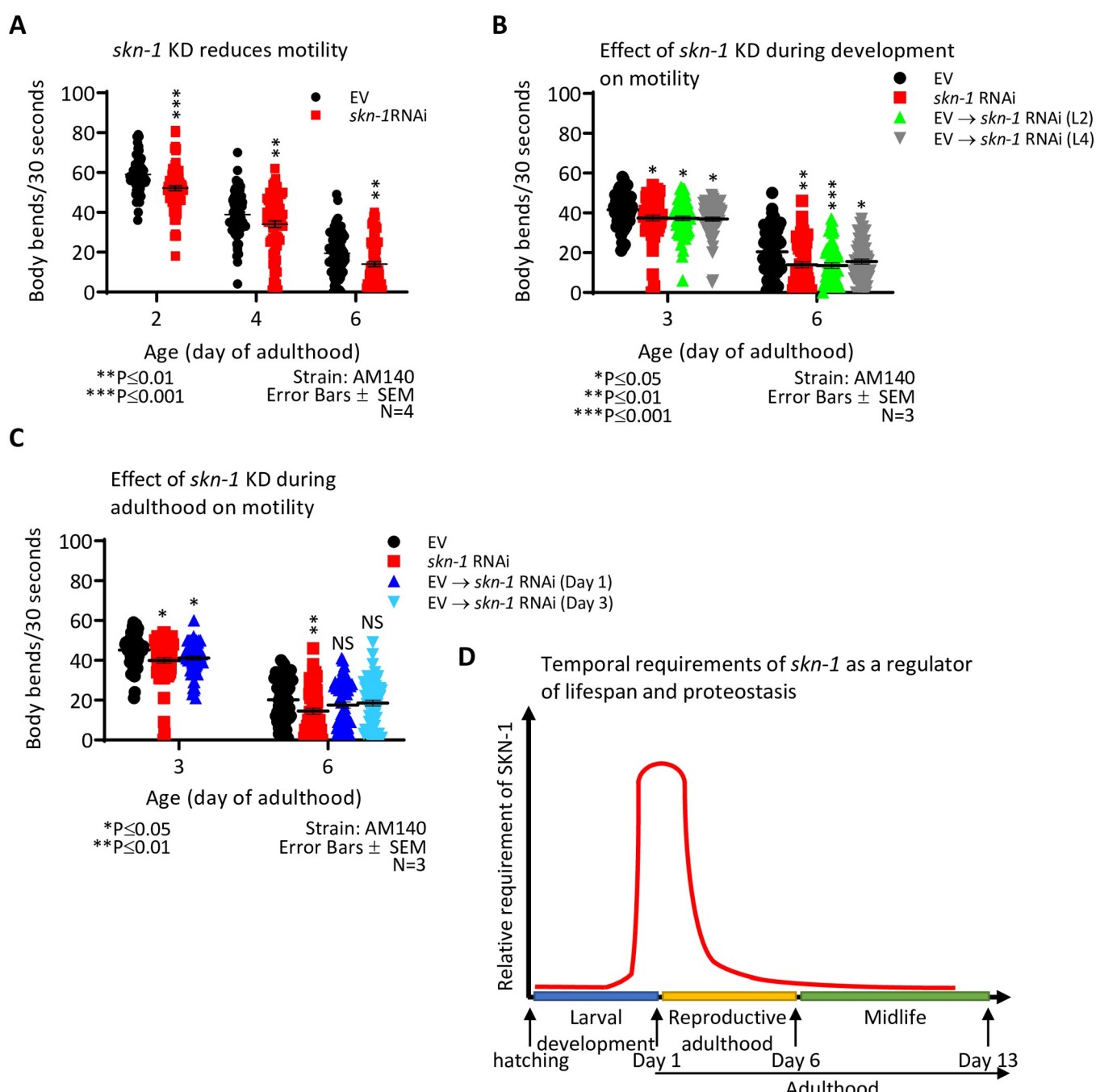

**Fig 3. *skn-1* expression is most critical from the late larval development through day 1 of adulthood to counter polyglutamine (polyQ) induced proteotoxicity.** (**A**) The knockdown of *skn-1* by RNAi significantly decreased the thrashing rates of worms that express polyQ35-YFP in their muscles (strain AM140) at day 2, 4 and 6 of adulthood (reduction of 11.64%, 12.29% and 27.60%, respectively). 80 worms were scored per treatment per timepoint. (**B**) Growing AM140 worms on control bacteria from hatching followed by transferring them onto *skn-1* RNAi at the L2 or L4 developmental stage resulted in enhanced proteotoxicity as measured by the thrashing assay (Reduction of 10.45% and 11.29% at day 3, respectively, and a reduction of 33.64% and 23.6% at day 6, respectively). 60 worms were scored per treatment per time point. (**C**) The knockdown of *skn-1* from day 1 (reduction of 10.38% and 7.35% when measured at day 3 and day 6, respectively) resulted in a significant trend of reduced motility only when measured on day 3 of adulthood. The transfer of EV treated worms to *skn-1* RNAi on day 3 of adulthood, resulted in a non-significant trend of reduced motility when measured on day 6 of adulthood (reduction of 10%). 60 worms were scored per treatment per time point. (**D**) A schematic illustration of the temporal requirements for SKN-1 as a regulator of lifespan and proteostasis.

the regulation of stress responses, such as oxidative stress [44], by SKN-1, reduces damage during early stages of life. Accordingly, the knockdown of *skn-1* during these early stages, in which the organism may be more vulnerable to metabolic insults, results in a less efficient activation of stress response mechanisms, higher rates of damage accumulation, and accelerates the process of aging.

An alternative model suggests that during late larval development through early adulthood, SKN-1 regulates the expression of genes whose products are needed for IIS reduction and DR to promote lifespan and proteostasis in adulthood. This theme may be supported by the observation that *skn-1* is highly expressed during the L2 larval stage (S6 Fig), a stage preceding the time window in which SKN-1 regulates lifespan and proteostasis, compared to the observed expression during adulthood of CF512 worms (S6 Fig). It is important to note that CF512 and wild type worms exhibit similar temporal requirements for *skn-1* as a lifespan determinant (Fig 1C and S2A and S2B Table). Similar requirements were also observed in lifespan experiments in which long-lived worms were used (Fig 1). It would be interesting to compare the gene networks that are differentially regulated by SKN-1 in late larval development through early adulthood to those which are regulated in other stages of life. Such target genes might encode constitutive heat shock proteins and inducible protective proteins. It would also be interesting to test whether SKN-1 and HSF-1 co-regulate target genes during development and whether the products of these genes are needed for the IIS to promote longevity and/or proteostasis during adulthood.

Another key question is where SKN-1 executes its longevity and proteostasis-promoting functions. Together, the known roles of neurons in the regulation of proteostasis [45,46], the prominent expression of *skn-1* in ASI neurons [36] and the differential regulation of DAF-16 and HSF-1 by a neuronal gene [38], suggest that the developmental functions of SKN-1 may be regulated at the organismal level by neurons. It would also be interesting to test whether the activity of signaling complexes that reside on caveolae, a membrane structures that we previously found to regulate Aβ-mediated proteotoxicity [47], is affected by the knockdown of *skn-1*. Further research is needed to test these possibilities.

An additional important aspect of the temporal analyses of IIS regulated transcription factors is the tight correlations between longevity and proteostasis. While DAF-16 regulates both lifespan and proteostasis during adulthood [33,35], and HSF-1 primarily during the L2 stage of larval development [34,35], SKN-1 govern these functions primarily from the late stages of larval development through early adulthood. This correlation supports the notion that the formation of an efficient proteostasis assurance mechanism is needed for IIS reduction, and perhaps also for DR, to slow the progression of aging and promote longevity [48].

The requirement of *skn-1* during early adulthood as a regulator of lifespan and proteostasis overlaps with the reproductive adulthood time window in which *daf-16* is needed to enable longevity of *daf-2* mutant worms [33]. It is tempting to speculate that DAF-16 and SKN-1 may co-regulate the expression of certain genes during reproductive adulthood. Indeed, SKN-1 and DAF-16 were shown to co-regulate the mitophagy mediator *dct-1* which promotes mitochondrial health [49]. Interestingly, two genes, *dod-17* and *dod-24* are regulated by SKN-1 and DAF-16 in opposing manners. While the knockdown of *daf-16* elevates the expression of these genes [50,51], the knockdown of *skn-1* reduces their expression levels [52]. These observations highlight the complex relations between aging-regulating pathways and their downstream transcription factors. The theory that aging-regulating pathways are coordinated is also supported by the observation that the IIS and proteostasis-maintaining signaling that originate from the reproductive system are linked at the post-translational level [53]. In accordance, it was reported recently that signals that originate from the reproductive system and from the IIS integrate to promote longevity [54].

## Supporting information

**S1 Fig. Analysis of *skn-1* RNAi efficiency by quantitative real-time PCR.**
(PDF)

**S2 Fig. The effects of *skn-1* RNAi on rates of paralysis of wild type animals and worms that express Aβ in their neurons.**
(PDF)

**S3 Fig. An illustration of the temporal knockdown of *skn-1* in Figs 1–3.**
(PDF)

**S4 Fig. Diluting *skn-1* RNAi expressing bacteria with EV bacteria has a minor effects on its efficiency as measured by quantitative real-time PCR.**
(PDF)

**S5 Fig. Thrashing assays show that the knockdown of *skn-1* by RNAi mildly reduces thrashing of wild-type worms at day 6 of adulthood and more prominently affects the thrashing rates of animals that express polyQ35-YFP in their neurons.**
(PDF)

**S6 Fig. A quantitative real-time PCR experiment to compare the *skn-1* expression levels in different stages of the nematode's lifecycle.**
(PDF)

**S1 Table. Numerical data of lifespan experiments presented at Fig 1A and 1B.**
(PDF)

**S2 Table.** A: Numerical data of a lifespan experiment presented at Fig 1C. B: Numerical data of a lifespan experiment of CF512 worms treated throughout life with EV or *skn-1* RNAi or transferred from EV bacteria onto *skn-1* RNAi at day 1 of adulthood.
(PDF)

**S3 Table.** A: Numerical data of a lifespan experiment presented at Fig 1D. B: Numerical data of a lifespan experiment of CB1370 worms treated throughout life with EV or *skn-1* RNAi or transferred from EV bacteria onto *skn-1* RNAi at day 1 of adulthood.
(PDF)

**S4 Table.** A: Numerical data of a lifespan experiment presented at Fig 1E. B: Numerical data of a lifespan experiment of DA1116 worms treated throughout life with EV or *skn-1* RNAi or transferred from EV bacteria onto *skn-1* RNAi at day 1 of adulthood.
(PDF)

## Acknowledgments

We thank all members of the Cohen laboratory for insightful discussions throughout the project.

## Author Contributions

**Conceptualization:** Danielle Grushko, Ehud Cohen.

**Data curation:** Danielle Grushko, Hana Boocholez, Amir Levine.

**Formal analysis:** Amir Levine.

**Funding acquisition:** Ehud Cohen.

**Methodology:** Ehud Cohen.

**Project administration:** Ehud Cohen.

**Supervision:** Ehud Cohen.

**Writing – original draft:** Ehud Cohen.

**Writing – review & editing:** Danielle Grushko, Hana Boocholez, Amir Levine.

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
