## [Decision Letter · Decision Letter 0]

31 Dec 2020

PONE-D-20-36759

Temporal Requirements of SKN-1/NRF as a Regulator of Lifespan and Proteostasis in Caenorhabditis elegans

PLOS ONE

Dear Dr. Cohen,

Thank you for submitting your manuscript to PLOS ONE. After careful consideration, we feel that it has merit but does not fully meet PLOS ONE’s publication criteria as it currently stands. Therefore, we invite you to submit a revised version of the manuscript that addresses the points raised during the review process.

Dear Author,

Kindly revise the manuscript as per the reviewers suggestisons.

We look forward to receiving your revised manuscript.

Kind regards,

Jamuna Subramaniam, Ph.D

Academic Editor

PLOS ONE

Journal Requirements:

2. Please include captions for your Supporting Information files at the end of your manuscript, and update any in-text citations to match accordingly. Please see our Supporting Information guidelines for more information: http://journals.plos.org/plosone/s/supporting-information

Additional Editor Comments:

Kindly revise the manuscript according to the Reviewers concerns.

Reviewers' comments:

Reviewer's Responses to Questions

**Comments to the Author**

1. Is the manuscript technically sound, and do the data support the conclusions?

Reviewer #1: Yes

Reviewer #2: Partly

2. Has the statistical analysis been performed appropriately and rigorously? 

Reviewer #1: Yes

Reviewer #2: No

3. Have the authors made all data underlying the findings in their manuscript fully available?

Reviewer #1: Yes

Reviewer #2: No

4. Is the manuscript presented in an intelligible fashion and written in standard English?

Reviewer #1: Yes

Reviewer #2: Yes

5. Review Comments to the Author

Reviewer #1: Several transcription factors are key regulators of longevity and proteostasis in C. elegans including SKN-1/Nrf, DAF-16/FOXO, and HSF-1. DAF-16 and HSF-1 have been shown to be required during certain stages of development and adulthood to regulate longevity, but a temporal analysis of SKN-1 has not previously been done. Grushko et al., use RNAi to temporally reduce SKN-1 activity to define the developmental or life stages in which it functions to regulate lifespan and proteostasis and discovered that SKN-1 is required in late larval development through young adult to regulate lifespan and proteostasis.

Major comments:

Although there is precedent for RNAi being used to determine the temporal requirements for daf-2, daf-16 and hsf-1, and the use of dcr-1 RNAi to temporally turn off the RNAi effect, there is always the inherent concern regarding the efficacy of RNAi to any specific gene. Expression levels and the stability of the protein may be determinants of how rapidly RNAi knockdown of a specific target occurs. The authors demonstrate that skn-1(RNAi) from hatching to day 1 adult (~48 hours) results in ~90% reduction in skn-1 mRNA. Thus, there may be up to a two-day delay from the time of initiation of RNAi to the reduction of skn-1 mRNA, and the timing of protein depletion may be longer. Ideally a time course of skn-1(RNAi) would have determined how quickly RNAi takes effect at the mRNA level, and how effective RNAi is at later life stages. Additionally, assaying SKN-1 protein levels should be considered, there are reported antibodies that could be used to test this. Barring these additional controls, the authors should at least address the limitations of their studies regarding the resolution of the timing requirements in the discussion.

Minor comments:

1) Last sentence of the introduction regarding that SKN-1 might regulate the expression of DAF-16 co-factors is intriguing, but is very speculative. I suggest removing or referencing expression data that would support this idea.

2) Strain nomenclature in the materials and methods and elsewhere is inconsistent. N2, not N2. Gene names and alleles in parentheses should be in italics. For example, eat-2(ad1116), please also provide the strain name (DA1116?). Indicate the genotypes of AGD1246, AM140 and AM1126 in the materials and methods.

3) In the methods for the proteotoxicity assays: “These 10 plates were randomly divided into 5 sets (2 plates, 24 worms per set).” It is unclear to me the purpose of this step.  Are you randomizing with the controls to reduce bias, or are the sets being analyzed by different people? Please clarify.

4) In the discussion neurons are considered as a potential site of SKN-1 action. However, neurons are refractory toward RNAi. Would not your results suggest that SKN-1 functions anywhere but neurons? This should be considered in the discussion.

5) In the discussion, it is speculated that SKN-1 and DAF-16 might co-regulate the expression of certain genes during reproductive adulthood. Oliveira et al (2009) Aging Cell PMID: 19575768 performed gene expression analysis with SKN-1 under various conditions and there are several studies on DAF-16 target genes, summarized well by Murphy (2006) Exp Gerontol PMID: 16934425. Seems that some data mining might identify candidates.

6) Last sentence of the Discussion is difficult to understand. These ideas need to be broken into two parts and stated more clearly.

7) Figure 2, why show panel A when the same data is being shown in panel B?

8) Although mentioned in the materials and methods, it is helpful to know the number of animals scored in figure legends.

Reviewer #2: This study, by Grushko et al., deals with the role of skn-1 transcription factor in C. elegans. Using a series of RNAi knockdown experiments, authors have concluded that skn-1 acts during the late larval and early adult stages to regulate the lifespan and proteostasis. The approach is essentially the same as described earlier for the hsf-1 study, albeit less in depth (Volovik et al., Aging Cell, 2012).

The current manuscript proposes that lifespan and proteostasis regulation depend on sequential and partially overlapping functions of skn-1 and two other transcription factors. While it is an interesting model, the data does not fully support it. More experiments are needed to strengthen the story. Specific comments are listed below.

Major comments

1. More details need to be provided in Methods on how animals were staged to obtain specific larval stage populations.

2. Stages of worms for various treatments and analyses are not very consistent. For example,

a. lifespans in wild-type background were analyzed after skn-1 knockdowns at L2, L4, adult day 1, adult day 5 and adult day 9 stages. But it was not the same for daf-2 and eat-2 mutants, making it hard to compare results.

b. Why did authors chose adult day 1 and day 3 stages for temporal analysis of skn-1 RNAi in AM140 worms (Fig. 3C, 3D) but day 2 and day 6 for others shown in Fig. 3A and 3B? Likewise, Sup 4B, C present data on day 4 and day 8 animals.

3. Some of the conclusions are confusing. For example, the authors claim that skn-1 is primarily needed later than the L4 larval stage to regulate lifespan (page 8). What is the evidence that skn-1 is not required during early larval (L1, L2 and L3) stages?

4. Paralysis phenotype is much stronger when skn-1 was knocked down in CL2006 animals starting the L2 stage compared to the L4 stage. Doesn’t this suggest that the gene plays a role during early larval stages?

5. Why are graphs 2A and 2B plotted differently? This is also the case for a few other figures. No explanation has been provided.

6. Authors have assumed that increased paralysis caused by skn-1 knockdown in aggregation-prone strains is due to increased proteotoxicity but did not provide any evidence for it. Could it be possible that skn-1 RNAi increases paralysis without affecting proteotoxicity?

7. Authors have argued that skn-1 mediates IIS pathway function by regulating downstream genes, but they have not demonstrated that skn-1 interacts with IIS signaling in their paralysis assays. They need to perform more experiments before making broad conclusions.

8. Sentences are confusing in many cases. For example,

a. ‘skn-1 RNAi throughout the experiment…’ on page 10 (used several times). What exactly does that mean? What duration is that?

b. On the same page, ’skn-1 plays no roles during development…’ is hard to understand.

c. The term ‘rate of paralysis’ has been used often. The data is, however, presented as a percentage of animals. The two are not the same.

8. While the number of batches of animals analyzed in some cases have been stated, this is not always the case. No information is provided in the Methods section either.

9. The efficiency of skn-1 RNAi was demonstrated just once at the beginning of the experiment. However, since many different stages of knockdowns were performed, additional evidence should be provided for the effectiveness of knockdowns. Also, does dcr-1 RNAi restore skn-1 levels?

10. Statistical analyses do not appear to be adequate, especially in cases where multiple paralysis phenotypes are being compared with controls. Consider using one-way ANOVA with Bonferroni correction or Tukey’s post hoc or Dunnett’s post hoc tests.

Other comments

- It is recommended that Results are divided into sub-sections to make it easier to follow different sets of findings.

- Manuscript should be carefully edited to eliminate all grammatical and sentence structure errors.

- Authors should consider providing a schematic to help understand the timings of various RNAi treatments and stages of analyses.

- Supplementary figure 4A title says 'Motility of neuronal polyQ35-YFP expressing worms', which seems to be incorrect.

6. PLOS authors have the option to publish the peer review history of their article (what does this mean?). If published, this will include your full peer review and any attached files.

Reviewer #1: No

Reviewer #2: No

---

## [Author Response · Author response to Decision Letter 0]

20 May 2021

A point-by-point rebuttal:

Reviewer #1: Several transcription factors are key regulators of longevity and proteostasis in C. elegans including SKN-1/Nrf, DAF-16/FOXO, and HSF-1. DAF-16 and HSF-1 have been shown to be required during certain stages of development and adulthood to regulate longevity, but a temporal analysis of SKN-1 has not previously been done. Grushko et al., use RNAi to temporally reduce SKN-1 activity to define the developmental or life stages in which it functions to regulate lifespan and proteostasis and discovered that SKN-1 is required in late larval development through young adult to regulate lifespan and proteostasis.

Major comments:

Although there is precedent for RNAi being used to determine the temporal requirements for daf-2, daf-16 and hsf-1, and the use of dcr-1 RNAi to temporally turn off the RNAi effect, there is always the inherent concern regarding the efficacy of RNAi to any specific gene. Expression levels and the stability of the protein may be determinants of how rapidly RNAi knockdown of a specific target occurs. The authors demonstrate that skn-1(RNAi) from hatching to day 1 adult (~48 hours) results in ~90% reduction in skn-1 mRNA. Thus, there may be up to a two-day delay from the time of initiation of RNAi to the reduction of skn-1 mRNA, and the timing of protein depletion may be longer. Ideally a time course of skn-1(RNAi) would have determined how quickly RNAi takes effect at the mRNA level, and how effective RNAi is at later life stages. 

We thank the referee for this important comment, conducted the experimental work to address it and modified the text accordingly. To test the relative knockdown efficiencies of skn-1 RNAi after exposure, we cultured CF512 worms on control bacteria (harboring the empty RNAi vector (EV)) and transferred groups of worms to plates that were seeded with skn-1 RNAi for 3, 6, 9, 12 or 24 hours prior to harvest. qPCR was used to assess the levels of skn-1 expression in all worm groups. We observed a gradual reduction in the levels of skn-1 mRNA as 12 hours of exposure reduced the expression of skn-1 by approximately 50% (revised supplemental figure S1C. N=4). Exposure of CF512 worms to skn-1 RNAi has led to a similar rate of reduction of ~55% (Fig. S1D of the revised manuscript).

We modified the text to describe these experiments and explain the limitations of RNAi-mediated conditional knockdown. Please see the beginning of the “results” section of the revised manuscript, pages 8 and top of page 9. In addition, we changed the discussion to claim that SKN-1 is needed at “late stages of development” instead of at L4. This better reflects the delay of ~12h in the reduction of skn-1 by RNAi (see for instance the first sentence of the “discussion” section at the bottom of page 14).

Additionally, assaying SKN-1 protein levels should be considered, there are reported antibodies that could be used to test this. Barring these additional controls, the authors should at least address the limitations of their studies regarding the resolution of the timing requirements in the discussion.

We appreciate this comment and made an effort to obtain SKN-1 antibodies, alas with no success. Unfortunately the sc-9244 SKN-1 antibody is discontinued and the company could not provide it.

Yet, we address the limitations of RNAi-mediated gene knockdown in the revised manuscript and mentioned the possible effects of protein stability (see page 9, first paragraph). 

In addition, the change in phenotype observed under conditions of skn-1 knockdown indicates that whilst protein stability might mask the full effects of skn-1 reduction, the reduction in skn-1 expression is enough to elicit phenotypical differences. 

Minor comments:

1) Last sentence of the introduction regarding that SKN-1 might regulate the expression of DAF-16 co-factors is intriguing, but is very speculative. I suggest removing or referencing expression data that would support this idea.

We agree, tuned down the sentence and added a reference (last sentence of the introduction, page 5 of the revised manuscript). 

2) Strain nomenclature in the materials and methods and elsewhere is inconsistent. N2, not N2. Gene names and alleles in parentheses should be in italics. For example, eat-2(ad1116), please also provide the strain name (DA1116?). Indicate the genotypes of AGD1246, AM140 and AM1126 in the materials and methods.

We have added the details regarding each strain as requested and corrected the nomenclature where it was needed throughout the manuscript.

3) In the methods for the proteotoxicity assays: “These 10 plates were randomly divided into 5 sets (2 plates, 24 worms per set).” It is unclear to me the purpose of this step. Are you randomizing with the controls to reduce bias, or are the sets being analyzed by different people? Please clarify.

We clarified this issue by explaining that it was done to prevent potential bias. Please see page 7, line of the revised manuscript.

4) In the discussion neurons are considered as a potential site of SKN-1 action. However, neurons are refractory toward RNAi. Would not your results suggest that SKN-1 functions anywhere but neurons? This should be considered in the discussion.

We agree with the referee that RNAi works less efficiently in neurons compared to other cell types. Nevertheless, the knockdown of neuronal genes can be achieved by RNAi. For instance, we have previously shown that RNAi towards the neuronal GPCR, gtr-1 renders the treated worm sensitive to heat stress (Maman et al., Journal of Neuroscience 2013). Thus, since the knockdown of skn-1 by RNAi results in clear phenotypes, this technique is an adequate method to conduct the temporal analysis that is described here. 

5) In the discussion, it is speculated that SKN-1 and DAF-16 might co-regulate the expression of certain genes during reproductive adulthood. Oliveira et al (2009) Aging Cell PMID: 19575768 performed gene expression analysis with SKN-1 under various conditions and there are several studies on DAF-16 target genes, summarized well by Murphy (2006) Exp Gerontol PMID: 16934425. Seems that some data mining might identify candidates.

We thank the referee for this comment and expanded the discussion to further elaborate on this this speculation. In the revised manuscript we provide more examples of genes that are regulated by both transcription factors: DAF-16 and SKN-1 (dod-17 and dod-24). We also added the relevant references. Please see page 17 of the revised manuscript.

6) Last sentence of the Discussion is difficult to understand. These ideas need to be broken into two parts and stated more clearly.

We revised the sentence as suggested and believe that is much clearer now.

7) Figure 2, why show panel A when the same data is being shown in panel B?

We modified figure 2 and all depicted panels display the sum of at least three independent experiments.

8) Although mentioned in the materials and methods, it is helpful to know the number of animals scored in figure legends.

We added the number of worms in each figure legend as suggested.

Reviewer #2: This study, by Grushko et al., deals with the role of skn-1 transcription factor in C. elegans. Using a series of RNAi knockdown experiments, authors have concluded that skn-1 acts during the late larval and early adult stages to regulate the lifespan and proteostasis. The approach is essentially the same as described earlier for the hsf-1 study, albeit less in depth (Volovik et al., Aging Cell, 2012).

The current manuscript proposes that lifespan and proteostasis regulation depend on sequential and partially overlapping functions of skn-1 and two other transcription factors. While it is an interesting model, the data does not fully support it. More experiments are needed to strengthen the story. Specific comments are listed below.

Major comments

1. More details need to be provided in Methods on how animals were staged to obtain specific larval stage populations.

We expanded the explanation of our synchronization protocol. Please see page 7 of the revised manuscript under “Lifespan assays”.

2. Stages of worms for various treatments and analyses are not very consistent. For example,

a. lifespans in wild-type background were analyzed after skn-1 knockdowns at L2, L4, adult day 1, adult day 5 and adult day 9 stages. But it was not the same for daf-2 and eat-2 mutants, making it hard to compare results.

The experiments that were conducted using e1370 or eat-2 animals were aimed to address the question of whether SKN-1 plays any role during development. Accordingly, we do not claim anything beyond the importance of SKN-1 for lifespan determination during development. We believe that our results sufficiently support these claims.

b. Why did authors chose adult day 1 and day 3 stages for temporal analysis of skn-1 RNAi in AM140 worms (Fig. 3C, 3D) but day 2 and day 6 for others shown in Fig. 3A and 3B? Likewise, Sup 4B, C present data on day 4 and day 8 animals.

We agree with this critique and improved this point in the revised manuscript. First, figure 3 has been modified. In the new figure 3A we present the relative rates of thrashing of untreated (EV) and skn-1 RNAi-treated AM140 worms at days 2, 4 and 6 of adulthood. We chose to test this at three ages since we did not know when the proteotoxic effect of knocking down skn-1would be apparent. Once we determined that day 2 old nematodes exhibit enhanced paralysis upon the knockdown of skn-1, we focused on two ages: day 2 and 6 of adulthood. 

In supplemental figure 5B of the revised manuscript (previously Fig S4, B and C), we used AM1126 worms that express polyQ35-YFP under the regulation of the rgef-1 neuronal promoter. The appearance of measurable phenotypes is different in distinct proteotoxicity strains. These differences emanate from various factors including the activity of the promoter and the site of integration of the exogenous gene. Thus, to observe a measurable phenotype we followed the worms at days 4 and 8.

We thank the referee for this comment and modified the text to better explain this issue (page 14, lines 6-7 from bottom). 

3. Some of the conclusions are confusing. For example, the authors claim that skn-1 is primarily needed later than the L4 larval stage to regulate lifespan (page 8). What is the evidence that skn-1 is not required during early larval (L1, L2 and L3) stages?

Since the knockdown of skn-1 from the L4 larval stage and on (Fig. 1A, gray line) and knocking it down throughout life (red line) result in a nearly identical lifespan shortening effects, we conclude that an uninterrupted expression of skn-1 during early development (prior to the application of RNAi) has no effect of lifespan determination. We clarified this issue in the revised manuscript (please see the bottom of page 9 and top of page 10). 

4. Paralysis phenotype is much stronger when skn-1 was knocked down in CL2006 animals starting the L2 stage compared to the L4 stage. Doesn’t this suggest that the gene plays a role during early larval stages?

The knockdown of skn-1 from the L2 stage and on (Fig. 2B, light green) and throughout life (2B, red) exhibit nearly identical effects of increased rates of paralysis. Therefore, we conclude the activity of SKN-1 in early developmental stages, between hatching and L2, has no role in protecting from Aβ-mediated proteotoxicity. In contrast, the knockdown of skn-1 from the L4 stage results in a partial restoration of the phenotype. This means that the activity of SKN-1 prior to L4 (when it was knocked down) is important for the worm to resist the toxicity of Aβ. Accordingly, we stated that SKN-1 is not needed during early development (up to the L2 stage) but is needed in later stages of development, for protection from proteotoxicity (page 11, second paragraph). 

Nevertheless, we further clarified this issue in page 11 of the revised manuscript.

5. Why are graphs 2A and 2B plotted differently? This is also the case for a few other figures. No explanation has been provided.

We agree and modified the results accordingly. In figure 2 of the revised manuscript all panels represent three or more independent repeats of the paralysis assay. 

6. Authors have assumed that increased paralysis caused by skn-1 knockdown in aggregation-prone strains is due to increased proteotoxicity but did not provide any evidence for it. Could it be possible that skn-1 RNAi increases paralysis without affecting proteotoxicity?

This a matter of definition. We refer to “proteotoxicity” as the damage that is elicited by toxic protein aggregation. Accordingly, reduction in motility is used here to measure proteotoxicity (please see Volovik et al., Methods 2014, for more details). We make no claim regarding the rate of protein aggregation as in this project we only used physiological assays. 

However, to better address this issue we present a control experiment (Fig. S2A) showing that the knockdown of skn-1 in wild type (N2) worms has only a marginal effect on the rate of paralysis. This minor effect is exclusively seen in the last day of the experiment – day 12 of adulthood.

7. Authors have argued that skn-1 mediates IIS pathway function by regulating downstream genes, but they have not demonstrated that skn-1 interacts with IIS signaling in their paralysis assays. They need to perform more experiments before making broad conclusions.

We agree with the referee and conducted an experiment to directly test whether the roles of SKN-1 as a regulator of proteostasis are associated with the IIS. To test this, we conducted a set of paralysis assays in which we treated CL2006 worms with a mixture of bacteria that express RNAi towards daf-2 and skn-1 (a dilution of skn-1 RNAi bacteria with EV bacteria efficiently knocks down the expression of skn-1 – Fig. S4). Our new results which are displayed as figure 2F of the revised manuscript, show that a concurrent knockdown of these two genes prevents daf-2 RNAi from providing its full protective effect from the paralysis phenotype. This observation supports the result of the involvement of SKN-1 in the protective mechanism that is acted upon the knockdown of daf-2. Alternatively, it may be possible that SKN-1 provides protection by a different mechanism. According to this notion, the paralysis rates that were observed in animals that were concurrently fed with skn-1 and daf-2 RNAi bacteria, is the sum of the protective effect of daf-2 RNAi and the deleterious outcome of knocking down skn-1.

We discussed this result at the discussion section of the revised manuscript (page 15, first paragraph). 

8. Sentences are confusing in many cases. For example,

a. ‘skn-1 RNAi throughout the experiment…’ on page 10 (used several times). What exactly does that mean? What duration is that?

We thank the referee for this comment and clarified the sentence. The worms that were subjected to the paralysis assay were followed up until day 12 of adulthood and thus, it is not accurate to claim that they were treated throughout life.

b. On the same page, ’skn-1 plays no roles during development…’ is hard to understand.

The sentence was modified to better clarify it. 

c. The term ‘rate of paralysis’ has been used often. The data is, however, presented as a percentage of animals. The two are not the same.

We thank the referee for this comment and clarified it throughout the manuscript. Now it appears as “the rate of paralysis within the worm population”.

8. While the number of batches of animals analyzed in some cases have been stated, this is not always the case. No information is provided in the Methods section either.

We apologize for not providing sufficient information and corrected this issue throughout the manuscript.

9. The efficiency of skn-1 RNAi was demonstrated just once at the beginning of the experiment. However, since many different stages of knockdowns were performed, additional evidence should be provided for the effectiveness of knockdowns. Also, does dcr-1 RNAi restore skn-1 levels?

We thank the referee for this important comment, conducted the experimental work to address it and modified the text accordingly. To test the relative knockdown efficiencies of skn-1 RNAi after exposure, we cultured CF512 worms on control bacteria (harboring the empty RNAi vector (EV)) and transferred groups of worms to plates that were seeded with skn-1 RNAi for 3, 6, 9, 12 or 24 hours prior to harvest. qPCR was used to assess the levels of skn-1 expression in all worm groups. We observed a gradual reduction in the levels of skn-1 mRNA as 12 hours of exposure reduced the expression of skn-1 by approximately 50% (revised supplemental figure S1C. N=4). Exposure of CF512 worms to skn-1 RNAi has led to a similar rate of reduction of ~55% (Fig. S1D of the revised manuscript).

We modified the text to describe these experiments and explain the limitations of RNAi-mediated conditional knockdown. Please see the beginning of the “results” section of the revised manuscript, pages 8 and top of page 9. In addition, we changed the discussion to claim that SKN-1 is needed at “late stages of development” instead of at L4. This better reflects the delay of ~12h in the reduction of skn-1 by RNAi (see for instance the first sentence of the “discussion” section at the bottom of page 14).

10. Statistical analyses do not appear to be adequate, especially in cases where multiple paralysis phenotypes are being compared with controls. Consider using one-way ANOVA with Bonferroni correction or Tukey’s post hoc or Dunnett’s post hoc tests.

We agree with the referee and re-analyzed the data using adequate methods: two way ANOVA followed by post hoc Holm-Šídák's corrections for multiple comparisons were used since in this statistical method the assumption is that each comparison is independent of the others (please see for reference: Steinbaugh et al. 2015 Elife. 2015 Aug 24;4:e07836. doi: 10.7554/eLife.07836). 

Other comments

- It is recommended that Results are divided into sub-sections to make it easier to follow different sets of findings.

We added sub-headings as suggested.

- Manuscript should be carefully edited to eliminate all grammatical and sentence structure errors.

We apologize for these errors and corrected them throughout the manuscript.

- Authors should consider providing a schematic to help understand the timings of various RNAi treatments and stages of analyses.

We added illustrations as suggested to figures 2 and 3. This illustration is displayed as supplemental figure 3A of the revised manuscript.

- Supplementary figure 4A title says 'Motility of neuronal polyQ35-YFP expressing worms', which seems to be incorrect.

We apologize for this mistake and corrected it in the revised version.

---

## [Decision Letter · Decision Letter 1]

14 Jun 2021

Temporal Requirements of SKN-1/NRF as a Regulator of Lifespan and Proteostasis in Caenorhabditis elegans

PONE-D-20-36759R1

Dear Dr. Cohen,

We’re pleased to inform you that your manuscript has been judged scientifically suitable for publication and will be formally accepted for publication once it meets all outstanding technical requirements.

Kind regards,

Jamuna Subramaniam, Ph.D

Academic Editor

PLOS ONE

Additional Editor Comments (optional):

I am glad to say that the manuscript will be accepted provided the minor revision required by Reviewer one is addressed satisfactorily.

Reviewers' comments:

Reviewer's Responses to Questions

**Comments to the Author**

1. If the authors have adequately addressed your comments raised in a previous round of review and you feel that this manuscript is now acceptable for publication, you may indicate that here to bypass the “Comments to the Author” section, enter your conflict of interest statement in the “Confidential to Editor” section, and submit your "Accept" recommendation.

Reviewer #1: (No Response)

Reviewer #2: All comments have been addressed

2. Is the manuscript technically sound, and do the data support the conclusions?

Reviewer #1: Yes

Reviewer #2: Yes

3. Has the statistical analysis been performed appropriately and rigorously? 

Reviewer #1: Yes

Reviewer #2: Yes

4. Have the authors made all data underlying the findings in their manuscript fully available?

Reviewer #1: Yes

Reviewer #2: Yes

5. Is the manuscript presented in an intelligible fashion and written in standard English?

Reviewer #1: Yes

Reviewer #2: Yes

6. Review Comments to the Author

Reviewer #1: Several transcription factors are key regulators of longevity and proteostasis in C. elegans including SKN-1/Nrf, DAF-16/FOXO, and HSF-1. DAF-16 and HSF-1 have been shown to be required during certain stages of development and adulthood to regulate longevity, but a temporal analysis of SKN-1 has not previously been done. Grushko et al., use RNAi to temporally reduce SKN-1 activity to define the developmental or life stages in which it functions to regulate lifespan and proteostasis and discovered that SKN-1 is required in late larval development through young adult to regulate lifespan and proteostasis.

Using RNAi to determine temporal gene requirements. My major concern with the first draft was the efficiency of skn-1 knockdown, how long this took so as to accurately assess the temporal requirements as mRNAs and their corresponding proteins can display varying degrees of stability. The authors make a good attempt to address this using qRT-PCR. They demonstrate that after animals fed skn-1 RNAi at hatching have a >90% depletion of skn-1 mRNA by day one of adulthood. Day one adult animals fed skn-1 RNAi for 12 or 24 hours results in ~50% reduction. Since many of their assays are over the course of adulthood it would have been nice to see if this level continues to go down, or if 50% reduction is the best achievable depletion by RNAi. It is unfortunate that they were not able to assess SKN-1 protein levels, as this makes it difficult to know the true temporal effects of skn-1 RNAi. It is proposed that there is a 12-hour lag time, but this could be much longer. It should be acknowledged as a caveat at the end of this section that the depletion of SKN-1 protein is not known. In the future, other methods for rapid depletion of proteins, like the AID system would be more effective than RNAi. Nevertheless, I find that the conclusions accurately reflect their results.

The last paragraph of the introduction continues to be too speculative. None of the experiments in this paper address the relationship between skn-1 and daf-16. The speculation that SKN-1 might regulate DAF-16 cofactors is over selling the paper. This should be left to the discussion. Something more general about the timing of SKN-1 requirements for lifespan and proteostasis relative to other TFs might be more appropriate.

Different lifespan requirements between N2 and CF512 strain.

Using N2 (wild-type) worms, the authors demonstrate that starting skn-1 RNAi feeding at the L4 larval stage resulted in a decrease in lifespan, but starting at day 1 did not significantly reduce lifespan. Without a particular rationale, these experiments were repeated using the CF512 strain. Here they find that skn-1 RNAi started at day 1 adult significantly suppressed lifespan. Why do you think there is a difference? The skn-1 efficacy experiments were also done in this strain and it is mentioned that it has similar lifespan as N2. But comparing the supplementary tables 1 and 2, it appears that in empty vector control, CF512 animals live 3.5-4 days longer than N2, which I suspect would be significantly different (CF512 is even more long lived than DA1116). In either case, the time difference between L4 and day 1 adult is not long, but the difference should be more prominently acknowledged.

Does skn-1 RNAi effects on paralysis stem from its role as an IIS component?

Please add a concluding sentence for the data in Figure 2F. My interpretation is that we don’t know from the experiment shown. Partial reduction of the daf-2 effect by loss of skn-1 could be consistent with skn-1 being one of the outputs of IIS signaling, but is also consistent with daf-2 and skn-1 functioning in independent, but opposite pathways. OK – this is discussed in the discussion, however a concluding sentence in the results would be good. I don’t think there is much added value in this experiment.

Continues to not follow C. elegans nomenclature

Although some changes were made, there is still many inconsistencies in the nomenclature. I did not intend for the eat-2 allele, ad1116, to be replaced throughout by the strain name DA1116 (but this is more in line with the rest of the paper), but that in general there seems to be confusion regarding what is an allele and what is a strain. In Figure 1, e1370 is referred to as a strain. The strain is CB1370, the allele is e1370. N2, not N2. Italics are not used consistently where needed. I don’t mean to be nit-picky, however I wouldn’t expect these types of nomenclature errors from an established C. elegans lab. It reflects a lack of attention to detail and suggests that the rest of the work might also be sloppy.

http://www.wormbook.org/chapters/www_nomenclature/caenornomenclature.html

Reviewer #2: Authors have addressed all of my concerns adequately. The revised manuscript is greatly improved. I support publication in PLOS ONE.

7. PLOS authors have the option to publish the peer review history of their article (what does this mean?). If published, this will include your full peer review and any attached files.

Reviewer #1: No

Reviewer #2: No

---

## [Editor Report · Acceptance letter]

22 Jun 2021

PONE-D-20-36759R1 

Temporal Requirements of SKN-1/NRF as a Regulator of Lifespan and Proteostasis in *Caenorhabditis elegans*

Dear Dr. Cohen:

I'm pleased to inform you that your manuscript has been deemed suitable for publication in PLOS ONE. Congratulations! Your manuscript is now with our production department. 

Kind regards, 

on behalf of

Dr. Jamuna Subramaniam 

Academic Editor

PLOS ONE